# Generalizing Information to the Evolution of Rational Belief

**DOI:** 10.3390/e22010108

**Published:** 2020-01-16

**Authors:** Jed A. Duersch, Thomas A. Catanach

**Affiliations:** Sandia National Laboratories, Livermore, CA 94550, USA

**Keywords:** information, Bayesian inference, entropy, self information, mutual information, Kullback–Leibler divergence, Lindley information, maximal uncertainty, proper utility

## Abstract

Information theory provides a mathematical foundation to measure uncertainty in belief. Belief is represented by a probability distribution that captures our understanding of an outcome’s plausibility. Information measures based on Shannon’s concept of entropy include realization information, Kullback–Leibler divergence, Lindley’s information in experiment, cross entropy, and mutual information. We derive a general theory of information from first principles that accounts for evolving belief and recovers all of these measures. Rather than simply gauging uncertainty, information is understood in this theory to measure change in belief. We may then regard entropy as the information we expect to gain upon realization of a discrete latent random variable. This theory of information is compatible with the Bayesian paradigm in which rational belief is updated as evidence becomes available. Furthermore, this theory admits novel measures of information with well-defined properties, which we explored in both analysis and experiment. This view of information illuminates the study of machine learning by allowing us to quantify information captured by a predictive model and distinguish it from residual information contained in training data. We gain related insights regarding feature selection, anomaly detection, and novel Bayesian approaches.

## 1. Introduction

This work integrates essential properties of information embedded within Shannon’s derivation of entropy [1] and the Bayesian perspective [2,3,4], which identifies probability with plausibility. We pursued this investigation in order to understand how to rigorously apply information-theoretic concepts to the theory of inference and machine learning. Specifically, we wanted to understand how to quantify the evolution of predictions given by machine learning models. Our findings are general, however, and bear implications for any situation in which states of belief are updated. We begin in Section 1.1 with an experiment that illustrates shortcomings with the way standard information measures would partition prediction information and residual information during machine learning training.

### 1.1. Shortcomings with Standard Approaches

Let us examine a typical MNIST [5] classifier. This dataset comprises a set of images of handwritten digits paired with labels. Let both *x* and *y* denote random variables corresponding, respectively, to an image and a label in a pair. In this situation, the training dataset contains independent realizations of such pairs from an unknown joint probability distribution. We would like to obtain a measurement of prediction information that quantifies a shift in belief from an uninformed initial state $q_{0}\left(y\right)$ to model predictions $q_{1}\left(y|x\right)$. The symmetric uninformed choice for $q_{0}\left(y\right)$ is uniform probability over all outcomes. Note that both $q_{0}\left(y\right)$ and $q_{1}\left(y|x\right)$ are simply hypothetical states of belief. Some architectures may approximate Bayesian inference, but we cannot always interpret these as the Bayesian prior and posterior.

Two measurements that are closely related to Shannon’s entropy are the Kullback–Leibler (KL) divergence [6,7] and Lindley’s information in experiment [8], which are computed respectively as
$$\begin{array}{cc} \; & D_{K\;L}\;\left[\;q_{1}\left(y|x\right)\;∥\;q_{0}\left(y\right)\;\right]=\;\int \;dy\;q_{1}\left(y|x\right)\log\;\left(\frac{q_{1}\left(y|x\right)}{q_{0}\left(y\right)}\right)\;\mathrm{and}\\ \; & D_{L}\;\left[\;q_{1}\left(y|x\right)\;∥\;q_{0}\left(y\right)\;\right]=\;\int \;dy\;q_{1}\left(y|x\right)\log\;\left(q_{1}\left(y|x\right)\right)-\;\int \;dy\;q_{0}\left(y\right)\log\;\left(q_{0}\left(y\right)\right). \end{array}$$

Whatever we choose, we would like to use a consistent construction to understand how much information remains unpredicted. After viewing a label outcome $\overset{ˇ}{y}$, we let $r(y|\overset{ˇ}{y})$ represent our new understanding of the actual state of affairs, which is a realization assigning full probability to the specified outcome. This distribution captures our most updated knowledge about *y*, and therefore constitutes rational belief. A consistent information measurement should then quantify residual information as the shift in belief from $q_{1}\left(y|x\right)$ to $r(y|\overset{ˇ}{y})$. For example, the KL version would be $D_{K\;L}\;\left[\;r\left(y|\overset{ˇ}{y}\right)\;∥\;q_{1}\left(y|x\right)\;\right]$.

In order to demonstrate shortcomings with each approach, some cases are deliberately mislabeled during model testing. We first compute information measurements while assuming the incorrect labels hold. Mislabeled cases are then corrected to $\hat{y}$ with corrected belief given by $r(y|\hat{y})$. This allows us to compare our first information measurements with corrected versions. An example of each belief state is shown in Figure 1 where the incorrect label **3** is changed to **0**.

Ideally, the sum of prediction information and residual information would be a conserved quantity, which would allow us to understand training as simply shifting information from a residual partition to the predicted partition. More importantly, however, prediction information should clearly capture prediction quality. Figure 2 shows information measurements corresponding to the example given.

Mislabeling and relabeling shows us that neither formulation of prediction information captures prediction quality. This is because these constructions simply have no affordance to account for our understanding of what is actually correct. Large KL residual information offers some indication of mislabeling and decreases when we correct the label, but total information is not conserved. As such, there is no intuitive notion for what appropriate prediction and residual information should be for a given problem. The Lindley formulation is substantially less satisfying. Although total information is conserved, neither metric changes and we have no indication of mislabeling.

The problem with these constructions is they do not recognize the gravity of the role of expectation. That is, reasonable expectation must be consistent with rational belief. We hold that our most justified understanding of what may be true provides a sound basis to measure changes in belief. Figure 3 gives a preview of information measurements in the framework of this theory. Total information, $\log_{2}\left(10\right)$ bits in this case, is conserved, and both prediction information and residual information react intuitively to mislabeling. Determining whether the predictive information is positive or negative provides a clear indication of whether the prediction was informative.

### 1.2. Our Contributions

In the course of pursuing a consistent framework in which information measurements may be understood, we have derived a theory of information from first principles that places all entropic information measures in a unified, interpretable context. By axiomatizing the properties of information we desire, we show that a unique formulation follows that subsumes critical properties of Shannon’s construction of entropy.

This theory fundamentally understands entropic information as a form of reasonable expectation that measures the change between hypothetical belief states. Expectation is not necessarily taken with respect to the distributions that represent the shift in belief, but rather with respect to a third distribution representing our understanding of what may actually be true. We found compelling foundations for this perspective within the Bayesian philosophy of probability as an extended logic for expressing and updating uncertainty [4,9]. Our understanding of what may be true, and therefore the basis for measuring information, should be rational belief. Rational belief [10,11,12,13,14] begins with probabilistically coherent prior knowledge, and is subsequently updated to account for observations using Bayes’ theorem. As a consequence, information associated with a change in belief is not a fixed quantity. Just as rational belief must evolve as new evidence becomes available, so also does the information we would reasonably assign to previous shifts in belief. By emphasizing the role of rational belief, this theory recognizes that the degree of validity we assign to past states of belief is both dynamic and potentially subjective as our state of knowledge matures.

As a consequence of enforcing consistency with rational belief, a second additivity property emerges; just as entropy can be summed over independent distributions, information gained over a sequence of observations can be summed over intermediate belief updates. Total information over such a sequence is independent of how results are grouped or ordered. This provides a compelling solution to the thought experiment above. Label information in training data is a conserved quantity and we motivate a formulation of prediction information that is directly tied to prediction quality.

Soofi, Ebrahimi, and others [15,16,17,18] identified key contributions to information theory in the decade following Shannon’s paper that are intrinsically tied to entropy. These are the Kullback–Leibler divergence, Lindley’s information in experiment, and Jaynes’ construction of entropy-maximizing distributions that are consistent with specified expectations. We show how this theory recovers these measures of information and admits new forms that may not have been previously associated with entropic information, such as the log pointwise posterior predictive measure of model accuracy [19]. We also show how this theory admits novel information-optimal probability distributions analogous to that of Jaynes’ maximum uncertainty. Having a consistent interpretation of information illuminates how it may be applied and what properties will hold in a given context. Moreover, this theoretical framework enables us to solve multiple challenges in Bayesian learning. For example, one such challenge is understanding how efficiently a given model incorporates new data. This theory provides bounds on the information gained by a model resulting from inference and allows us to characterize the information provided by individual observations.

The rest of this paper is organized as follows. Section 2 discusses notation and background regarding entropic information, Bayesian inference, and reasonable expectation. Section 3 contains postulates that express properties of information we desire, the formulation of information that follows, and other related measures of information. Section 4 analyzes general consequences and properties of this formulation. Section 5 discusses further implications with respect to Bayesian inference and machine learning. Section 6 explores negative information with computational experiments that illustrate when it occurs, how it may be understood, and why it is useful. Section 7 summarizes these results and offers a brief discussion of future work. Appendix A proves our principal result. Appendix B contains all corollary proofs. Appendix C provides key computations used in experiments.

## 2. Background and Notation

Shannon’s construction of entropy [1] shares a fundamental connection with thermodynamics. The motivation is to facilitate analysis of complex systems which can be decomposed into independent subsystems. The essential idea is simple—when probabilities multiply, entropy adds. This abstraction allows us to compose uncertainties across independent sources by simply adding results. Shannon applied this perspective to streams of symbols called channels. The number of possible outcomes grows exponentially with the length of a symbol sequence, whereas entropy grows linearly. This facilitates a rigorous formulation of the rate of information conveyed by a channel and analysis of what is possible in the presence of noise.

The property of independent additivity is used in standard training practices for machine learning. Just as thermodynamic systems and streams of symbols break apart, so does an ensemble of predictions over independent observations. This allows us to partition training sets into batches and compute cross-entropy [20] averages. MacKay [21] gives a comprehensive discussion of information in the context of learning algorithms. Tishby [22] examines information trends during neural network training.

A second critical property of entropy, which is implied by Shannon and further articulated by both Barnard [23] and Rényi [24], is that entropy is an expectation. Given a latent random variable *z*, we denote the probability distribution over outcomes as p(z). Stated as an expectation, entropy is defined as
$$S\;\left[\;p(z)\;\right]=\;\int \;dz\;p\left(z\right)\log\;\left(\frac{1}{p(z)}\right)=E_{p(z)}\log\;\left(\frac{1}{p(z)}\right).$$

Following Shannon, investigators developed a progression of divergence measures between general probability distributions, $q_{0}\left(z\right)$ and $q_{1}\left(z\right)$. Notable cases include the Kullback–Leibler divergence, Rényi’s information of order-α [24], and Csiszár’s *f*-divergence [25]. Ebrahimi, Soofi, and Soyer [18] offered an examination of these axiomatic foundations and generalizations with a primary focus on entropy and the KL divergence. Recent work on axiomatic foundations for generalized entropies [26] includes constructions that are suitable for strongly-interacting systems [27] and axiomatic derivations of other forms of entropy, including Sharma–-Mittal and Frank-–Daffertshofer entropies [28]. Further work uses group theory to relate properties of systems to corresponding notions of entropy and correlation laws [29].

### 2.1. Bayesian Reasoning

The Bayesian view of probability, going back to Laplace [2] and championed by Jeffreys [3] and Jaynes [4], focuses on capturing our beliefs. This perspective considers a probability distribution as an abstraction that attempts to model these beliefs. This view subsumes all potential sources of uncertainty and provides a comprehensive scope that facilitates analysis in diverse contexts.

In the Bayesian framework, the prior distribution p(z) expresses initial beliefs about some latent variable *z*. Statisticians, scientists, and engineers often have well-founded views about real-world systems that form the basis for priors. Examples include physically realistic ranges of model parameters or plausible responses of a dynamical system. In the case of total ignorance, one applies the principle of insufficient reason [30]—we should not break symmetries of belief without justification. Jaynes’ construction of maximally uncertainty distributions [31] generalizes this principle, which we discuss further in Section 4.6.

As observations *x* become available, we update belief from the prior distribution to obtain the posterior distribution p(z|x), which incorporates this new knowledge. This update is achieved by applying Bayes’ theorem
$$p\left(z|x\right)=\frac{p(x|z)p(z)}{p(x)}\;\mathrm{where}\;p\left(x\right)=\;\int \;dz\;p\left(x|z\right)p\left(z\right).$$

The likelihood distribution p(x|z) expresses the probability of observations given any specified value of *z*. The normalization constant p(x) is also the probability of *x* given the prior belief that has been specified. Within Bayesian inference, this is also called model evidence and it is used to evaluate a model structure’s plausibility for generating the observations.

Shore and Johnson [32,33] provide an axiomatic foundation for updating belief that recovers the principles of maximum entropy and minimum cross-entropy when prior evidence consists of known expectations. For reference, we summarize these axioms as

*Uniqueness*. When belief is updated with new observations, the result should be unique.*Coordinate invariance*. Belief updates should be invariant to arbitrary choices of coordinates.*System independence*. The theory should yield consistent results when independent random variables are treated either separately or jointly.*Subset independence*. When we partion potential outcomes into disjoint subsets, the belief update corresponding to conditioning on subset membership first should yield the same result as updating first and conditioning on the subset second.

Jizba and Korbel [34] investigated generalizations of entropy for which the maximum entropy principle satisfies these axioms.

Integrating the maturing notion of belief found within the Bayesian framework with information theory recognizes that our perception of how informative observations are depends on how our beliefs develop, which is dynamic as our state of knowledge grows.

### 2.2. Probability Notation

Random variables are denoted in boldface; for example, *x*. Typically, *x* and *y* will imply observable measurements and *z* will indicate either a latent explanatory variable or unknown observable. Each random variable is implicitly associated with a corresponding probability space, including the set of all possible outcomes $\Omega_{z}$, a σ-algebra $F_{z}$ of measurable subsets, and a probability measure $P_{z}$ which maps subsets of events to probabilities. We then express the probability measure as a distribution function p(z).

A realization, or specific outcome, will be denoted with either a check $\overset{ˇ}{z}$, or, for discrete distributions only, a subscript $z_{i}$, where i∈[n] and [n]={1,2,…,n}. If it is necessary to emphasize the value of a distribution at a specific point or realization, we will use the notation $p(z=\overset{ˇ}{z})$. Conditional dependence is denoted in the usual fashion as p(z|x). The joint distribution is then p(x,z)=p(z|x)p(x), and marginalization is obtained by p(x)=∫dzp(x,z). When two distributions are equivalent over all subsets of nonzero measure, we use notation $q_{0}\left(z\right)\equiv p\left(z\right)$ or $q_{1}\left(z\right)\equiv p\left(z|x\right)$.

The probability measure allows us to compute expectations over functions f(z) which are denoted
$$E_{p(z)}f\left(z\right)=\;\int \;dz\;p\left(z\right)f\left(z\right).$$

The support of integration or summation is implied to be the same as the support of p(z); that is, the set of outcomes for which p(z)>0. For example, in both the discrete case above and continuous cases, such as a distribution on the unit interval $z\in R_{\left[0,1\right]}$, the integral notation should be interpreted respectively as
$$\;\int \;dz\;p\left(z\right)f\left(z\right)=\sum_{i=1}^{n}p\left(z=z_{i}\right)f\left(z_{i}\right)\;\mathrm{and}\;\;\int \;dz\;p\left(z\right)f\left(z\right)=\;\int_{0}^{1}\;\;d\overset{ˇ}{z}\;p\left(z=\overset{ˇ}{z}\right)f\left(\overset{ˇ}{z}\right).$$

### 2.3. Reasonable Expectation and Rational Belief

The postulates and theory in this work concern the measurement of a shift in belief from an initial state $q_{0}\left(z\right)$ to an updated state $q_{1}\left(z\right)$. In principle, these are any hypothetical states of belief. For example, they could be predictions given by the computational model in Section 1.1, previous beliefs held before observing additional data, or convenient approximations of a more informed state of belief. A third state r(z), *rational belief*, serves a distinct role as the distribution over which expectation is taken. When we wish to emphasize this role, we also refer to r(z) as the *view of expectation*.

To understand the significance of rational belief, we briefly review work by Cox [9] regarding reasonable expectation from two perspectives on the meaning of probability. The first perspective understands probability as a description of relative frequencies in an ensemble. If we prepare a large ensemble of independent random variables, $Z=\{z_{i}\;|\;i\in \left[n\right]\}$, and each is realized from a proper (normalized) probability distribution p(z), then the relative frequency of outcomes within each subset $\omega \in \Omega_{z}$ will approach the probability measure $P_{z}\left(\omega \right)$ for large *n*. It follows that the ensemble mean of any transformation f(z) will approach the expectation
$$\lim_{n\to \infty }\frac{1}{n}\sum_{i=1}^{n}f\left(z_{i}\right)=E_{p(z)}f\left(z\right).$$

The difficulty arises when we distinguish *what is true* from *what may be known*, given limited evidence. This falls within the purview of the second perspective, the Bayesian view, regarding probability as an extended logic. To illustrate, suppose *z* is the value of an unknown real mathematical constant. The true probability distribution would be a Dirac delta $p\left(z\right)\equiv \delta \left(z-\overset{ˇ}{z}\right)$ assigning unit probability to the unknown value $\overset{ˇ}{z}$. Accordingly, each element in the ensemble above would take the same unknown value. If we have incomplete knowledge r(z) regarding the distribution of plausible values, then we can still compute an expectation $E_{r(z)}f\left(z\right)$, but we must bear in mind that the result only approximates the unknown true expectation. Since the expectation is limited by the credibility of r(z), we seek to drive belief towards the truth as efficiently as possible from available evidence to fulfill this role.

Within Bayesian Epistemology, rational belief is defined as a belief that is unsusceptible to a Dutch Book. When an agent’s beliefs correspond to their willingness to places bets, a Dutch Book [11,12,13,14,35] means that it is possible for a bookie to construct a table of bets that the agent finds acceptable but also guarantees that the agent will lose money. Therefore the existence of such a table corresponds to the agent holding an irrational state of belief. When multiple bets are allowed to be conditioned on a sequence of outcomes, it has been shown that the agent must use Bayes’ Theorem to account for previous outcomes in the sequence to update beliefs regarding subsequent outcomes to avoid irrationality [14].

For our purposes, it is sufficient to say that if we have a coherent prior belief in a latent variable and a likelihood function that implies beliefs about observations, Bayes’ theorem incorporates observational evidence to from the posterior distribution representing rational belief. For example, we could measure inference information from prior belief $q_{0}\left(z\right)\equiv p\left(z\right)$ to the first posterior $q_{1}\left(z\right)\equiv p\left(z|x\right)$ conditioned on an observation *x*. When we have additional evidence *y* that complements *x*, then rational belief must corresponds to a second inference r(z)≡p(z|x,y), because retaining the belief $q_{1}\left(z\right)$ would not account for *y*. Likewise, if *z* is an observable realization, then rational belief must assign full probability to the observed outcome $\overset{ˇ}{z}$. This case is specifically denoted as $r(z|\overset{ˇ}{z})$, and in continuous settings it is equivalent to the Dirac delta function $r\left(z|\overset{ˇ}{z}\right)\equiv \delta \left(z-\overset{ˇ}{z}\right)$.

### 2.4. Remarks on Bayesian Objectivism and Subjectivism

Within the Bayesian philosophy, we may disagree about whether or not rational belief is unique. This disagreement corresponds to objectivist versus subjectivist views of Bayesian epistemology; see [4,36] for a discussion. Note that this is not the same as the more general view of objective versus subjective probabilities.

In the objectivist’s view, one’s beliefs must be consistent with the entirety of evidence and prior knowledge must be justified by sound principles of reason. Therefore, anyone with the same body of evidence must hold the same rational belief. In contrast, the subjectivist holds that one’s prior beliefs do not need justification. Provided evidence is taken into account using Bayes’ theorem, the resulting posterior is rational for any prior as long as the prior is coherent. Note that the subjectivist view does not imply that all beliefs are equally valid. It simply allows validity in the construction of prior belief to be derived from other notions of utility, such as computational feasibility.

While the following postulates in Section 3 and derivation of Theorem 1 do not require adoption of either perspective, these philosophies influence how we understand reasonable expectation. The objective philosophy implies that an information measurement is justified to the same degree as the view of expectation that defines it, whereas the subjective philosophy entertains information analysis with any view of expectation.

## 3. Information and Evolution of Belief

In order to provide context for comparison, we begin by presenting the properties of entropic information originally put forward by Shannon using our notation.

### 3.1. Shannon’s Properties of Entropy

Given a discrete probability distribution p(z) for which $z\in \{z_{i}\;|\;i\in \left[n\right]\}$, the entropy $S\;\left[\;p(z)\;\right]$ is continuous in the probability of each outcome $p(z=z_{i})$.If all outcomes are equally probable, namely, $p(z=z_{i})=1/n$, then $S\;\left[\;p(z)\;\right]$ is monotonically increasing in *n*.The entropy of a joint random variable $S\;\left[\;p(z,w)\;\right]$ can be decomposed using a chain rule expressing conditional dependence
$$S\;\left[\;p(z,w)\;\right]=S\;\left[\;p(z)\;\right]+E_{p(z)}S\;\left[\;p(w|z)\;\right].$$

The first point is aimed at extending Shannon’s derivation, which employs rational probabilities, to real-valued probabilities. The second point drives at understanding entropy as a measure of uncertainty; as the number of possible outcomes increases, each realization becomes less predictable. This results in entropy taking positive values. The third point is critical, as—not only does it encode independent additivity, it implies that entropic information is computed as an expectation.

We note that Fadeeve [37] gives a simplified set of postulates. Rènyi [24] generalizes information by replacing the last point with a weaker version, which simply requires independent additivity, but not conditional expectation. This results in α-divergences. Csiszàr [25] generalizes this further using convex functions *f* to obtain *f*-divergences.

### 3.2. Postulates

Rather than repeating direct analogs of Shannon’s properties in the context of evolving belief, it is both simpler and more illuminating to be immediately forthcoming regarding the key requirement of information in the perspective of this theory.

**Postulate** **1.**
*Entropic information associated with the change in belief from $q_{0}\left(z\right)$ to $q_{1}\left(z\right)$ is quantified as an expectation over belief r(z), which we call the view of expectation. As an expectation, it must have the functional form*
$$I_{r(z)}\;\left[\;q_{1}\left(z\right)\;∥\;q_{0}\left(z\right)\;\right]=\;\int \;dz\;r\left(z\right)f\;\left(r\left(z\right),q_{1}\left(z\right),q_{0}\left(z\right)\right).$$


**Postulate** **2.**
*Entropic information is additive over independent belief processes. Taking joint distributions associated with two independent random variables z and w to be $q_{0}\left(z,w\right)=q_{0}\left(z\right)q_{0}\left(w\right)$, $q_{1}\left(z,w\right)=q_{1}\left(z\right)q_{1}\left(w\right)$, and r(z,w)=r(z)r(w) gives*
$$I_{r(z)r(w)}\;\left[\;q_{1}\left(z\right)q_{1}\left(w\right)\;∥\;q_{0}\left(z\right)q_{0}\left(w\right)\;\right]=I_{r(z)}\;\left[\;q_{1}\left(z\right)\;∥\;q_{0}\left(z\right)\;\right]+I_{r(w)}\;\left[\;q_{1}\left(w\right)\;∥\;q_{0}\left(w\right)\;\right].$$


**Postulate** **3.**
*If belief does not change then no information is gained, regardless of the view of expectation,*
$$I_{r(z)}\;\left[\;q_{0}\left(z\right)\;∥\;q_{0}\left(z\right)\;\right]=0.$$


**Postulate** **4.**
*The information gained from any normalized prior state of belief $q_{0}\left(z\right)$ to an updated state of belief r(z) in the view of r(z) must be nonnegative*
$$I_{r(z)}\;\left[\;r\left(z\right)\;∥\;q_{0}\left(z\right)\;\right]\geq 0.$$


The first postulate requires information to be reassessed as belief changes. The most justified state of belief, based on the entirety of observations, will correspond to the most justified view of information. The second postulate is the additive form of Shore and Johnson’s Axiom 3, system independence. That is, we need some law of composition, addition in this case, that allows independent random variables to be treated separately and arrive at the same result as treating them jointly.

By combining the first two postulates, it is possible to show that $f\left(r,q,p\right)=\log\left(r^{\gamma }q^{\alpha }p^{\beta }\right)$ for constants α,β,γ. See Appendix A for details. The third postulate constrains these exponential constants and the fourth simply sets the sign of information.

### 3.3. Principal Result

**Theorem** **1.**
***Information as a Measure of Change in Belief.***
*Information measurements that satisfy these postulates must take the form*
$$I_{r(z)}\;\left[\;q_{1}\left(z\right)\;∥\;q_{0}\left(z\right)\;\right]=\alpha \;\int \;dz\;r\left(z\right)\log\left(\frac{q_{1}\left(z\right)}{q_{0}\left(z\right)}\right)\;for\;some\;\alpha >0.$$


Proof is given in Appendix A. As Shannon notes regarding entropy, α corresponds to a choice of units. Typical choices are natural units α=1 and bits $\alpha =\log{\left(2\right)}^{-1}$. We employ natural units in analysis and bits in experiments.

Although it would be possible to combine Postulate 1 and Postulate 2 into an analog of Shannon’s chain rule as a single postulate, doing so would obscure the reasoning behind the construction. We leave the analogous chain rule as a consequence in Corollary 1. Regarding Shannon’s proof that entropy is the only construction that satisfies properties he provides, we observe that he has restricted attention to functionals acting upon a single distribution. The interpretation of entropy is discussed in Section 4.1.

Normalization of r(z) is a key property of rational belief and reasonable expectation. As for $q_{0}\left(z\right)$ and $q_{1}\left(z\right)$, however, nothing postulated prevents analysis respecting improper or non-normalizable probability distributions. In the Bayesian context, such distributions merely represent relative plausibility among subsets of outcomes. We caution that such analysis is a further abstraction, which requires additional care for consistent interpretation.

We remark that although Rènyi and Csiszàr were able to generalize divergence measures by weakening Shannon’s chain rule to independent additivity, inclusion of the first postulate prevents such generalizations. We suspect, however, that if we replace Postulate 1 with an alternative functional that incorporates rational belief into information measurements, or we replace Postulate 2 with an alternative formulation of system independence, then other compelling information theories would follow.

### 3.4. Regarding the Support of Expectation

The proof given assumes $q_{0}\left(z\right)$ and $q_{1}\left(z\right)$ take positive values over the support of the integral, which is also the support of r(z). In the Bayesian context, we also have
$$q_{1}\left(z\right)=\frac{p\left(x|z\right)q_{0}\left(z\right)}{p(x)}\;\mathrm{and}\;r\left(z\right)=\frac{p\left(y|z,x\right)q_{1}\left(z\right)}{p(y|x)}.$$

Accordingly, if for some $\overset{ˇ}{z}$ we have $q_{1}\left(\overset{ˇ}{z}\right)=0$, it follows that $r(\overset{ˇ}{z})=0$. Likewise, $q_{0}\left(\overset{ˇ}{z}\right)=0$ would imply both $q_{1}\left(\overset{ˇ}{z}\right)=0$ and $r(\overset{ˇ}{z})=0$. This forbids information contributions that fall beyond the scope of the proof. Even so, the resulting form is analytic and admits analytic continuation.

Since both $\lim_{\varepsilon \to 0}\left[\varepsilon \log\varepsilon \right]=0$ and $\lim_{\varepsilon \to 0}\left[\varepsilon \log\varepsilon^{-1}\right]=0$, limits of information of the form
$$\lim_{\varepsilon \to 0}\varepsilon \log\left(\frac{q_{1}}{q_{0}}\right),\;\lim_{\varepsilon \to 0}\varepsilon \log\left(\frac{q_{1}}{\varepsilon }\right),\;\lim_{\varepsilon \to 0}\varepsilon \log\left(\frac{\varepsilon }{q_{0}}\right),\;\mathrm{and}\;\lim_{\varepsilon \to 0}\varepsilon \log\left(\frac{\varepsilon }{\varepsilon }\right)$$
are consistent with restricting the domain of integration (or summation) to the support of r(z). We gain further insight by considering limits of the form
$$\lim_{\varepsilon \to 0}r\log\left(\frac{\varepsilon }{q}\right)\;\mathrm{and}\;\lim_{\varepsilon \to 0}r\log\left(\frac{q}{\varepsilon }\right).$$

Information diverges to −∞ in the first case and +∞ in the second. This is consistent with the fact that no finite amount of data will recover belief over a subset that has been strictly forbidden from consideration, which bears ramifications for how we understand rational belief.

If belief is not subject to influence from evidence, it is difficult to credibly construe an inferred outcome as having rationally accounted for that evidence. Lindley calls this Cromwell’s rule [38]; we should not eliminate a potential outcome from consideration unless it is logically false. The principle of insufficient reason goes further by avoiding unjustified creation of information that is not influenced by evidence.

### 3.5. Information Density

The Radon–Nikodym theorem [39] formalizes the notion of density that relates two measures. If we assign both probability and a second measure to any subset within a probability space, then there exists a density function, unique up to subsets of measure zero, such that the second measure is equivalent to the integral of said density over any subset.

**Definition** **1.**
***Information Density.***
*We take the Radon–Nikodym derivative to obtain information density of the change in belief from $q_{0}\left(z\right)$ to $q_{1}\left(z\right)$.*
$$D\;\left[\;q_{1}\left(z\right)\;∥\;q_{0}\left(z\right)\;\right]=\frac{dI_{r(z)}\;\left[\;q_{1}\left(z\right)\;∥\;q_{0}\left(z\right)\;\right]}{dr(z)}=\log\;\left(\frac{q_{1}\left(z\right)}{q_{0}\left(z\right)}\right).$$


The key property we find in this construction is independence from the view of expectation. As such, information density encodes all potential information outcomes one could obtain from this theory. Furthermore, this formulation is amenable to analysis of improper distributions. For example, it proves useful to consider information density corresponding to constant unit probability density $q_{1}\left(z\right)\equiv 1$, which is discussed further in Section 4.1.

### 3.6. Information Pseudometrics

The following pseudometrics admit interpretations as notions of distance between belief states that remain compatible with Postulate 1. This is achieved by simply taking the view of expectation r(z) to be the weight function in weighted-$L^{p}$ norms of information density. These constructions then satisfy useful properties of pseudometrics:*Positivity*, $L_{r(z)}^{p}\;\left[\;q_{1}\left(z\right)\;∥\;q_{0}\left(z\right)\;\right]\geq 0$; *Symmetry*, $L_{r(z)}^{p}\;\left[\;q_{1}\left(z\right)\;∥\;q_{0}\left(z\right)\;\right]=L_{r(z)}^{p}\;\left[\;q_{0}\left(z\right)\;∥\;q_{1}\left(z\right)\;\right]$; *Triangle inequality*,
$$L_{r(z)}^{p}\;\left[\;q_{2}\left(z\right)\;∥\;q_{0}\left(z\right)\;\right]\leq L_{r(z)}^{p}\;\left[\;q_{2}\left(z\right)\;∥\;q_{1}\left(z\right)\;\right]+L_{r(z)}^{p}\;\left[\;q_{1}\left(z\right)\;∥\;q_{0}\left(z\right)\;\right].$$

**Definition** **2.**
**$L^{p}$ Information Pseudometrics.**
*We may construct pseudometrics that measure distance between states of belief $q_{0}\left(z\right)$ and $q_{1}\left(z\right)$ with the view of expectation r(z), by taking weighted-$L^{p}$ norms of information density where the view of expectation serves as the weight function*
$$L_{r(z)}^{p}\;\left[\;q_{1}\left(z\right)\;∥\;q_{0}\left(z\right)\;\right]={\left(\;\int \;dz\;r\left(z\right){\left|\log\;\left(\frac{q_{1}\left(z\right)}{q_{0}\left(z\right)}\right)\right|}^{p}\right)}^{1/p}\;for\;some\;p\geq 1.$$


Note that taking p=1 results in a pseudometric that is also a pure expectation. The *homogeneity* property of seminorms, ∥αx∥=|α|∥x∥ for α∈R, implies that these constructions retain the units of measure of information density; if information density is measured in bits, these distances have units of bits as well. Symmetry is obvious from inspection, and the other properties follow by construction as seminorms. Specifically, positivity follows from the fact that ${\left|\cdot \right|}^{p}$ is a convex function for p≥1. The lower bound immediately follows from Jensen’s inequality: $$L_{r(z)}^{p}\;\left[\;q_{1}\left(z\right)\;∥\;q_{0}\left(z\right)\;\right]\geq \left|I_{r(z)}\;\left[\;q_{1}\left(z\right)\;∥\;q_{0}\left(z\right)\;\right]\right|\geq 0.$$

A short proof of the triangle inequality is given in Appendix B.

We observe that if $q_{0}\left(z\right)$ and $q_{1}\left(z\right)$ are measurably distinct over the support of r(z), then the measured distance must be greater than zero. We may regard states of belief $q_{0}\left(z\right)$ and $q_{1}\left(z\right)$ as weakly equivalent in the view of r(z) if their difference is immeasurable over the support of r(z). That is, if $q_{0}\left(z\right)$ and $q_{1}\left(z\right)$ only differ over subsets of outcomes that are deemed by r(z) to be beyond plausible consideration, then in the view of r(z), they are equivalent. As such, these pseudometrics could be regarded as subjective metrics in the view of r(z). The natural definition of information variance also satisfies the properties of a pseudometric and is easily interpreted as a standard statistical construct.

**Definition** **3.**
***Information Variance.***
*Information variance between belief states $q_{0}\left(z\right)$ and $q_{1}\left(z\right)$ in the view of expectation r(z) is simply the variance of information density*
$${\mathrm{Var}}_{r(z)}\;\left[\;q_{1}\left(z\right)\;∥\;q_{0}\left(z\right)\;\right]=\;\int \;dz\;r\left(z\right){\left(\log\;\left(\frac{q_{1}\left(z\right)}{q_{0}\left(z\right)}\right)-\varphi \right)}^{2},$$
*where $\varphi =I_{r(z)}\;\left[\;q_{1}\left(z\right)\;∥\;q_{0}\left(z\right)\;\right]$.*


## 4. Corollaries and Interpretations

The following corollaries examine primary consequences of Theorem 1. Note that multiple random variables may be expressed as a single joint variable such as $z=\left(z_{1},z_{2},\ldots ,z_{n}\right)$. The following corollaries explore one or two components at a time, such as variables $z_{1}$ and $z_{2}$ or observations *x* and *y*. Extensions to multiple random variables easily follow.

Note that the standard formulation of conditional dependence holds for all probability distributions in Corollary 1. That is, given an arbitrary joint distribution $q(z_{1},z_{2})$, we can compute the marginalization as $q\left(z_{1}\right)\equiv \;\int \;dz_{2}\;q\left(z_{1},z_{2}\right)$ and conditional dependence follows by the Radon–Nikodym derivative to obtain $q\left(z_{2}|z_{1}\right)\equiv \frac{q(z_{1},z_{2})}{q(z_{1})}$. All proofs are contained in Appendix B.

**Corollary** **1.**
***Chain Rule of Conditional Dependence.***
*Information associated with joint variables decomposes as*
$$\begin{array}{cc} I_{r(z_{1},z_{2})}\;\left[\;q_{1}\left(z_{1},z_{2}\right)\;∥\;q_{0}\left(z_{1},z_{2}\right)\;\right] & =I_{r(z_{1})}\;\left[\;q_{1}\left(z_{1}\right)\;∥\;q_{0}\left(z_{1}\right)\;\right]\\ \; & +E_{r(z_{1})}I_{r(z_{2}|z_{1})}\;\left[\;q_{1}\left(z_{2}|z_{1}\right)\;∥\;q_{0}\left(z_{2}|z_{1}\right)\;\right]. \end{array}$$


**Corollary** **2.**
***Additivity Over Belief Sequences.**
Information gained over a sequence of belief updates is additive within the same view. Given initial belief $q_{0}\left(z\right)$, intermediate states $q_{1}\left(z\right)$ and $q_{2}\left(z\right)$, and the view r(z), we have*
$$I_{r(z)}\;\left[\;q_{2}\left(z\right)\;∥\;q_{0}\left(z\right)\;\right]=I_{r(z)}\;\left[\;q_{2}\left(z\right)\;∥\;q_{1}\left(z\right)\;\right]+I_{r(z)}\;\left[\;q_{1}\left(z\right)\;∥\;q_{0}\left(z\right)\;\right].$$


**Corollary** **3.**
***Antisymmetry.**
Information from $q_{1}\left(z\right)$ to $q_{0}\left(z\right)$ is the negative of information from $q_{0}\left(z\right)$ to $q_{1}\left(z\right)$*
$$I_{r(z)}\;\left[\;q_{0}\left(z\right)\;∥\;q_{1}\left(z\right)\;\right]=-I_{r(z)}\;\left[\;q_{1}\left(z\right)\;∥\;q_{0}\left(z\right)\;\right].$$


### 4.1. Entropy

Shannon’s formalization of entropy as uncertainty may be consistently understood as the expectation of information gained by realization. We first reconstruct information contained in realization. We then define the general form of entropy in the discrete case, which is cross entropy, and finally, the standard form of entropy follows.

**Corollary** **4.**
***Realization Information (Discrete).**
Let z be a discrete random variable $z\in \{z_{i}\;|\;i\in \left[n\right]\}$. Information gained by realization $\overset{ˇ}{z}$ from q(z) in the view of realization $r(z|\overset{ˇ}{z})$ is*
$$I_{r(z|\overset{ˇ}{z})}\;\left[\;r\left(z|\overset{ˇ}{z}\right)\;∥\;q\left(z\right)\;\right]=D\;\left[\;1\;∥\;q(z=\overset{ˇ}{z})\;\right].$$


**Corollary** **5.**
***Cross Entropy (Discrete).**
Let z be a discrete random variable $z\in \{z_{i}\;|\;i\in \left[n\right]\}$ and $\overset{ˇ}{z}$ be a hypothetical realization. Expectation over the view $r(\overset{ˇ}{z})$ of information gained by realization from belief q(z) recovers cross entropy*
$$E_{r(\overset{ˇ}{z})}I_{r(z|\overset{ˇ}{z})}\;\left[\;r\left(z|\overset{ˇ}{z}\right)\;∥\;q\left(z\right)\;\right]=I_{r(z)}\;\left[\;1\;∥\;q(z)\;\right]=S_{r(z)}\;\left[\;q(z)\;\right].$$


**Corollary** **6.**
***Entropy (Discrete).**
Let z be a discrete random variable $z\in \{z_{i}\;|\;i\in \left[n\right]\}$ and $\overset{ˇ}{z}$ be a hypothetical realization. Expectation over plausible realizations $q(\overset{ˇ}{z})$ of information gained by realization from belief q(z) recovers entropy*
$$E_{q(\overset{ˇ}{z})}I_{r(z|\overset{ˇ}{z})}\;\left[\;1\;∥\;q(z)\;\right]=I_{q(z)}\;\left[\;1\;∥\;q(z)\;\right]=S\;\left[\;q(z)\;\right].$$


Shannon proved that this is the only construction as a functional acting on a single distribution q(z) that satisfies his properties. As mentioned earlier, the information notation $I_{q(z)}\;\left[\;1\;∥\;q(z)\;\right]$ requires some subtlety of interpretation. Probability density 1 over all discrete outcomes $z\in \Omega_{z}$ is not generally normalized. Although these formulas are convenient abstractions that result from formal derivations as expectations in the discrete case, nothing prevents us from applying them in continuous settings, which recovers the typical definitions in such cases.

In the continuous setting, we must emphasize that this definition of entropy is not consistent with taking the limit of a sequence of discrete distributions that converges in probability density to a continuous limiting distribution. The entropy of such a sequence diverges to infinity, which matches our intuition; the number of bits required to specify a continuous (real) random variable also diverges.

### 4.2. Information in an Observation

As discussed in Section 2.3, we may regard $q_{0}\left(z\right)\equiv p\left(z\right)$ as prior belief and $q_{1}\left(z\right)\equiv p\left(z|x\right)$ as the posterior conditioned on the observation of *x*. Without any additional evidence, we must hold r(z)≡p(z|x) to be rational belief and we recover the Kullback–Liebler divergence as the rational measure of information gained by the observation of *x*, but with a caveat: once we obtain additional evidence *y*, then information in the observation of *x* must be recomputed as $I_{p(z|x,y)}\;\left[\;p(z|x)\;∥\;p(z)\;\right]$. In contrast, this theory holds that Lindley’s corresponding measure,
$$D_{L}\;\left[\;p(z|x)\;∥\;p(z)\;\right]=S\;\left[\;p(z)\;\right]-S\;\left[\;p(z|x)\;\right],$$
is not the information gained by the observation of *x*; it is simply the difference in uncertainty before and after the observation.

### 4.3. Potential Information

We now consider expectations over hypothetical future observations *w* that would influence belief in *z* as a latent variable. Given belief p(z), the probability of an observation *w* is p(w)=∫dzp(w|z)p(z) as usual.

**Corollary** **7.**
***Consistent Future Expectation.**
Let the view p(z) express present belief in the latent variable z and w represent a future observation. The expectation over plausible w of information in the belief-shift from $q_{0}\left(z\right)$ to $q_{1}\left(z\right)$ in the view of rational future belief p(z|w) is equal to information in the present view*
$$E_{p(w)}I_{p(z|w)}\;\left[\;q_{1}\left(z\right)\;∥\;q_{0}\left(z\right)\;\right]=I_{p(z)}\;\left[\;q_{1}\left(z\right)\;∥\;q_{0}\left(z\right)\;\right].$$


**Corollary** **8.**
***Mutual Information.**
Let the view p(z) express present belief in the latent variable z and w represent a future observation. Expectation of information gained by a future observation w is mutual information*
$$E_{p(w)}I_{p(z|w)}\;\left[\;p(z|w)\;∥\;p(z)\;\right]=I_{p(z,w)}\;\left[\;p(z,w)\;∥\;p(z)p(w)\;\right].$$


**Corollary** **9.**
***Realization Limit.**
Let z be a latent variable and $\overset{ˇ}{z}$ be the limit of increasing observations to obtain arbitrary precision over plausible values of z. Information gained from $q_{0}\left(z\right)$ to $q_{1}\left(z\right)$ in the realization limit $r(z|\overset{ˇ}{z})$ is pointwise information density*
$$I_{r(z|\overset{ˇ}{z})}\;\left[\;q_{1}\left(z\right)\;∥\;q_{0}\left(z\right)\;\right]=D\;\left[\;q_{1}\left(z=\overset{ˇ}{z}\right)\;∥\;q_{0}\left(z=\overset{ˇ}{z}\right)\;\right].$$


### 4.4. Consistent Optimization Analysis

Bernardo [40] shows that integrating entropy-like information measures with Bayesian inference provides a logical foundation for rational experimental design. He considers potential utility functions, or objectives for optimization, which are formulated as kernels of expectation over posterior belief updated by the outcome of an experiment. Bernardo then distinguishes the belief a scientist reports from belief that is justified by inference.

For a utility function to be *proper*, the Bayesian posterior must be the unique optimizer of expected utility over all potentially reported beliefs. In other words, a proper utility function must not provide an incentive to lie. His analysis shows that Lindley information is a proper utility function. Corollary 10 holds that information in this theory also provides proper utility. Thus, information measures are not simply ad hoc objectives; they facilitate consistent optimization-based analysis that recovers rational belief.

**Corollary** **10.**
***Information Is a Proper Utility Function.**
Taking the rational view p(z|x) over the latent variable z conditioned upon an experimental outcome x, the information $I_{p(z|x)}\;\left[\;q(z)\;∥\;p(z)\;\right]$ from prior belief p(z) to reported belief q(z) is a proper utility function. That is, the unique optimizer recovers rational belief*
$$q^{*}\;\left(z\right)\equiv \underset{q(z)}{argmax}\;I_{p(z|x)}\;\left[\;q(z)\;∥\;p(z)\;\right]\equiv p\left(z|x\right).$$


We would like to go a step further and show that when information from $q_{0}\left(z\right)$ to $q_{1}\left(z\right)$ is positive in the view of r(z), we may claim that $q_{1}\left(z\right)$ is closer to r(z) than $q_{0}\left(z\right)$. For this claim to be consistent, we must show that any perturbation that unambiguously drives belief $q_{1}\left(z\right)$ toward the view r(z) must also increase information. The complementary perturbation response with respect to $q_{0}\left(z\right)$ immediately follows by Corollary 3.

**Corollary** **11.**
***Proper Perturbation Response.**
Let $q_{1}\left(z\right)$ be measurably distinct from the view r(z) and $I_{r(z)}\;\left[\;q_{1}\left(z\right)\;∥\;q_{0}\left(z\right)\;\right]$ be finite. Let the perturbation η(z) preserve normalization and drive belief toward r(z) on all measurable subsets. It follows that*
$$\lim_{\varepsilon \to 0}\frac{\partial }{\partial \varepsilon }I_{r(z)}\;\left[\;q_{1}\left(z\right)+\varepsilon \eta \left(z\right)\;∥\;q_{0}\left(z\right)\;\right]>0.$$


It bears repeating, by Corollary 8, that mutual information captures expected proper utility, which provides a basis for rational experimental design and feature selection.

### 4.5. Discrepancy Functions

Ebrahimi, Soofi, and Soyer [18] discuss information discrepancy functions, which have two key properties. First, a discrepancy function is nonnegative $D\;\left[\;q_{1}\left(z\right)\;∥\;q_{0}\left(z\right)\;\right]\geq 0$ with equality if and only if $q_{1}\left(z\right)\equiv q_{0}\left(z\right)$. Second, if we hold $q_{0}\left(z\right)$ fixed, then $D\;\left[\;q_{1}\left(z\right)\;∥\;q_{0}\left(z\right)\;\right]$ is convex in $q_{1}\left(z\right)$. One of the reasons information discrepancy functions are useful is that they serve to identify independence. Random variables *x* and *z* are independent if and only if p(x)≡p(x|z). Therefore, we have $D\;\left[\;p(x,z)\;∥\;p(x)p(z)\;\right]\geq 0$ with equality if and only if *x* and *z* are independent, noting that p(x,z)≡p(x|z)p(z). This has implications regarding sensible generalizations of mutual information.

Theorem 1 does not satisfy information discrepancy properties unless the view of expectation is taken to be $r\left(z\right)\equiv q_{1}\left(z\right)$, which is the KL divergence. We note, however, that information pseudometrics and information variance given in Section 3.6 satisfy a weakened formulation. Specifically, $L_{r(z)}^{p}\;\left[\;q_{1}\left(z\right)\;∥\;q_{0}\left(z\right)\;\right]\geq 0$ with equality if and only if $q_{0}\left(z\right)$ and $q_{1}\left(z\right)$ are weakly equivalent in the view of r(z). Likewise, these formulations are convex in information density $D\;\left[\;q_{1}\left(z\right)\;∥\;q_{0}\left(z\right)\;\right]$.

### 4.6. Jaynes Maximal Uncertainty

Jaynes uses entropy to analytically construct a unique probability distribution for which uncertainty is maximal while maintaining consistency with a specified set of expectations. This construction avoids unjustified creation of information and places the principle of insufficient reason into an analytic framework within which the notion of symmetry generalizes to informational symmetries conditioned upon observed expectations.

We review how Jaynes constructs the resulting distribution $r^{*}\;\left(z\right)$. Let such kernels of expectation be denoted $f_{i}\left(z\right)$ for i∈[n] and the observed expectations be $E_{r(z)}\;\left[f_{i}\left(z\right)\right]=\varphi_{i}$. The objective of optimization is
$$r^{*}\;\left(z\right)=\underset{r(z)}{\mathrm{argmax}}S\;\left[\;r(z)\;\right]\;\mathrm{subject}\;\mathrm{to}\;E_{r(z)}f_{i}\left(z\right)=\varphi_{i}\;\forall \;i\in \left[n\right].$$

The Lagrangian, which captures both the uncertainty objective and expectation constraints, is
$$L\;\left[\;r(z),\lambda \;\right]=\;\int \;dz\;r\left(z\right)\left(\log\;\left(\frac{1}{r(z)}\right)-\sum_{i=1}^{n}\lambda_{i}\left(f_{i}\left(z\right)-\varphi_{i}\right)\right),$$
where $\lambda \in R^{n}$ is the vector of Lagrange multipliers. This Lagrangian formulation satisfies the variational principle in both r(z) and λ. Variational analysis yields the optimizer
$$r^{*}\;\left(z\right)\propto \exp\;\left(\sum_{i=1}^{n}\lambda_{i}f_{i}\left(z\right)\right).$$

#### Information-Critical Distributions

Rather than maximizing entropy, we may minimize $I_{r(z)}\;\left[\;r\left(z\right)\;∥\;q_{0}\left(z\right)\;\right]$ while maintaining consistency with specified expectations. Since the following corollary holds for general distributions $q_{0}\left(z\right)$, including the improper case $q_{0}\left(z\right)\equiv 1$, this includes Jaynes’ maximal uncertainty as a minimization of negative entropy.

**Corollary** **12.**
***Minimal Information.** Given kernels of expectation $f_{i}\left(z\right)$ and specified expectations $E_{r(z)}\;\left[f_{i}\left(z\right)\right]=\varphi_{i}$ for i∈[n], the distribution $r^{*}\;\left(z\right)$ that satisfies these constraints while minimizing information $I_{r(z)}\;\left[\;r\left(z\right)\;∥\;q_{0}\left(z\right)\;\right]$ is given by*
$$r^{*}\;\left(z\right)\propto q_{0}\left(z\right)\exp\;\left(\sum_{i=1}^{n}\lambda_{i}f_{i}\left(z\right)\right)\;for\;some\;\lambda \in R^{n}.$$


### 4.7. Remarks on Fisher Information

Fisher provides an analytic framework to assess the suitability of a pointwise latent description of a probability distribution [41]. As Kullback and Leibler note, the functional properties of information in Fisher’s construction are quite different from Shannon’s, and thus, we do not regard Fisher information as a form of entropic information. Fisher’s construction, however, can be rederived and understood within this theory. He begins with the assumption that there is some latent realization $\overset{ˇ}{z}$ for which $p(x|\overset{ˇ}{z})$ is an exact description of the true distribution of *x*. We can then define the *Fisher score* as the gradient of information from any independent prior belief $q_{0}\left(x\right)$ to a pointwise latent description p(x|z), in the view $p(x|\overset{ˇ}{z})$
$$f=\nabla_{z}I_{p(x|\overset{ˇ}{z})}\;\left[\;p\left(x|z\right)\;∥\;q_{0}\left(x\right)\;\right].$$

Note that $\overset{ˇ}{z}$ is fixed by assumption, despite remaining unknown. By the variational principle, the score must vanish at the optimizer $z^{*}\;$. By Corollary 10, the optimizer must be $z^{*}\;=\overset{ˇ}{z}$. We can then assess the sensitivity of information to the parameter *z* at the optimizer $z^{*}\;$ by computing the Hessian. This recovers an equivalent construction of the Fisher matrix within this theory: $$F_{ij}=\frac{\partial^{2}}{\partial z_{i}\partial z_{j}}I_{p(x|\overset{ˇ}{z})}\;\left[\;p\left(x|z\right)\;∥\;q_{0}\left(x\right)\;\right].$$

The primary idea behind this construction is that high-curvature in *z* implies that a pointwise description is both suitable and a well-conditioned optimization problem.

#### Generalized Fisher Matrix

We may eliminate the assumption of an exact pointwise description and generalize analogous formulations to arbitrary views of expectation.

**Definition** **4.**
***Generalized Fisher Score.** Let r(x) be the view of expectation regarding an observable x. The gradient with respect to z of information from independent prior belief $q_{0}\left(x\right)$ to a pointwise description p(x|z) gives the score*
$$f=\nabla_{z}I_{r(x)}\;\left[\;p\left(x|z\right)\;∥\;q_{0}\left(x\right)\;\right].$$


**Definition** **5.**
***Generalized Fisher Matrix.** Let r(x) be the view of expectation regarding an observable x. The Hessian matrix with respect to components of z of information from independent prior belief $q_{0}\left(x\right)$ to the pointwise description p(x|z) gives the generalized Fisher matrix*
$$F_{ij}=\frac{\partial^{2}}{\partial z_{i}\partial z_{j}}I_{r(x)}\;\left[\;p\left(x|z\right)\;∥\;q_{0}\left(x\right)\;\right].$$


Again, a local optimizer $z^{*}\;$ must satisfy the variational principle and yield a score of zero. The generalized Fisher matrix would typically be evaluated at such an optimizer $z^{*}\;$.

## 5. Information in Inference and Machine Learning

We now examine model information and predictive information provided by inference. Once we have defined these information measurements, we derive upper and lower bounds between them that we anticipate being useful for future work. Finally, we show how inference information may be constrained, which addresses some challenges in Bayesian inference.

### 5.1. Machine Learning Information

Akaike [42] first introduced information-based complexity criteria as a strategy for model selection. These ideas were further developed by Schwarz, Burnham, and Gelman [19,43,44]. We anticipate these notions will prove useful in future work to both understand and control the problem of memorization in machine learning training. Accordingly, we discuss how this theory views model complexity and distinguishes formulations of predictive information and residual information.

In machine learning, observations correspond to matched pairs of inputs and labels $Y=\left\{\left(x^{\left(j\right)},y^{\left(j\right)}\right)\;|\;j\in \left[T\right]\right\}$. For each sample *j* of *T* training examples, we would like to map the input $x^{\left(j\right)}$ to an output label $y^{\left(j\right)}$. Latent variables θ are unknown model parameters from a specified model family or computational structure. A *model* refers to a specific parameter state and the predictions that the model computes are $p(y^{\left(j\right)}|x^{\left(j\right)},\theta )$. Since the definitions and derivations that follow hold with respect to either single cases or the entire training ensemble, we will use shorthand notation p(y|θ) to refer to both scenarios.

We denote the initial state of belief in model parameters as $q_{0}\left(\theta \right)$ and update belief during training as $q_{i}\left(\theta \right)$ for i∈[n]. We can then compute predictions from any state of model belief by marginalization $q_{i}\left(y\right)\equiv \;\int \;d\theta \;p\left(y|\theta \right)q_{i}\left(\theta \right)$.

**Definition** **6.**
***Model Information.**
Model information from initial belief $q_{0}\left(\theta \right)$ to updated belief $q_{i}\left(\theta \right)$ in the view of r(θ) is given by*
$$I_{r(\theta )}\;\left[\;q_{i}\left(\theta \right)\;∥\;q_{0}\left(\theta \right)\;\right]\;for\;i\in \left[n\right].$$


When we compute information contained in training labels, the label data obviously provide the rational view. This is represented succinctly by $r(y|\overset{ˇ}{y})$, which assigns full probability to specified outcomes. Again, if we need to be explicit, then this could be written as $r(y|x^{\left(j\right)},y^{\left(j\right)})$ for each case in the training set.

**Definition** **7.**
***Predictive Label Information.**
The realization of training labels is the rational view $r(y|\overset{ˇ}{y})$ of label plausibility. We compute information from prior predictive belief $q_{0}\left(y\right)$ to predictive belief $q_{i}\left(y\right)$ in this view as*
$$I_{r(y|\overset{ˇ}{y})}\;\left[\;q_{i}\left(y\right)\;∥\;q_{0}\left(y\right)\;\right].$$


In the continuous setting, this formulation is closely related to *log pointwise predictive density* [19]. We can also define complementary label information that is not contained in the predictive model.

**Definition** **8.**
***Residual Label Information (Discrete).**
Residual information in the label realization view $r(y|\overset{ˇ}{y})$ is computed as*
$$I_{r(y|\overset{ˇ}{y})}\;\left[\;r\left(y|\overset{ˇ}{y}\right)\;∥\;q_{i}\left(y\right)\;\right].$$


Residual information is equivalent to cross-entropy if the labels are full realizations. We note, however, that if training labels are probabilistic and leave some uncertainty, then replacing both occurrences of $r(y|\overset{ˇ}{y})$ above with a general distribution r(y) would correctly calibrate residual information, so that if predictions were to match label distributions, then residual information would be zero.

As a consequence of Corollary 2, the sum of predictive label information and residual label information is always constant. This allows us to rigorously frame predictive label information as a fraction of the total information contained in training labels. Moreover, Corollary 11 assures us that model perturbations that drive predictive belief toward the label view must increase predictive information. In the continuous setting, just as the limiting form of entropy discussed in Section 4.1 diverges, so too does residual information diverge. Predictive label information, however, remains a finite alternative. This satisfies our initial incentive for this investigation.

There is a second type of predictive information we may rationally construct, however. Rather than considering predictive information with respect to specified label outcomes, we might be interested in the information we expect to obtain about new samples from the generative process. If we regard marginalized predictions $q_{i}\left(y\right)$ as our best approximation of this process, then we would simply measure change in predictive belief in this view.

**Definition** **9.**
***Predictive Generative Approximation.**
We may approximate the distribution of new outcomes from model belief $q_{i}\left(\theta \right)$ using the predictive marginalization $q_{i}\left(y\right)\equiv \;\int \;d\theta \;p\left(y|\theta \right)q_{i}\left(\theta \right)$. If we hold this to be the rational view of new outcomes from the generative process, predictive information is*
$$I_{q_{i}\left(y\right)}\;\left[\;q_{i}\left(y\right)\;∥\;q_{0}\left(y\right)\;\right].$$


### 5.2. Inference Information Bounds

In Bayesian inference, we have prior belief in model parameters $q_{0}\left(\theta \right)\equiv p\left(\theta \right)$ and the posterior inferred from training data $q_{1}\left(\theta \right)\equiv p\left(\theta |\overset{ˇ}{y}\right)$. The predictive marginalizations are called the *prior predictive* and *posterior predictive* distributions respectively: $$p\left(y\right)\equiv \;\int \;d\theta \;p\left(y|\theta \right)p\left(\theta \right)\;\mathrm{and}\;p\left(y|\overset{ˇ}{y}\right)\equiv \;\int \;d\theta \;p\left(y|\theta \right)p\left(\theta |\overset{ˇ}{y}\right).$$

We can derive inference information bounds for Bayesian networks [45]. Let *y*, $\theta_{1}$, and $\theta_{2}$ represent a directed graph of latent variables. In general, the joint distribution can always be written as $p\left(y,\theta_{1},\theta_{2}\right)=p\left(\theta_{2}|\theta_{1},y\right)p\left(\theta_{1}|y\right)p\left(y\right)$. The property of *local conditionality* [46] means $p\left(\theta_{2}|\theta_{1},\overset{ˇ}{y}\right)\equiv p\left(\theta_{2}|\theta_{1}\right)$. That is, belief dependence in $\theta_{2}$ is totally determined by that of $\theta_{1}$ just as belief in $\theta_{1}$ is computed from $\overset{ˇ}{y}$.

**Corollary** **13.**
***Joint Local Inference Information.**
Inference information in $\theta_{1}$ gained by having observed $\overset{ˇ}{y}$ is equivalent to the inference information in both $\theta_{1}$ and $\theta_{2}$.*
$$I_{p(\theta_{1},\theta_{2}|\overset{ˇ}{y})}\;\left[\;p\left(\theta_{1},\theta_{2}|\overset{ˇ}{y}\right)\;∥\;p\left(\theta_{1},\theta_{2}\right)\;\right]=I_{p(\theta_{1}|\overset{ˇ}{y})}\;\left[\;p\left(\theta_{1}|\overset{ˇ}{y}\right)\;∥\;p\left(\theta_{1}\right)\;\right].$$


**Corollary** **14.**
***Monotonically Decreasing Local Inference Information.**
Inference information in $\theta_{2}$ gained by having observed $\overset{ˇ}{y}$ is bound above by inference information in $\theta_{1}$.*
$$I_{p(\theta_{2}|\overset{ˇ}{y})}\;\left[\;p\left(\theta_{2}|\overset{ˇ}{y}\right)\;∥\;p\left(\theta_{2}\right)\;\right]\leq I_{p(\theta_{1}|\overset{ˇ}{y})}\;\left[\;p\left(\theta_{1}|\overset{ˇ}{y}\right)\;∥\;p\left(\theta_{1}\right)\;\right].$$


This shows that an inference yields nonincreasing information as we compound the inference with locally conditioned latent variables, which is relevant for sequential predictive computational models, such as neural networks. We observe that the inference sequence from training data $\overset{ˇ}{y}$ to model parameters θ, to new predictions *y*, is also a locally conditioned sequence. If belief in a given latent variable is represented as a probability distribution, this places bounds on what transformations are compatible with the progression of information. For example, accuracy measures which snap the maximum probability outcome of a neural network to unit probability impose an unjustified creation of information.

**Corollary** **15.**
***Inferred Information Upper Bound.**
Model information in the posterior view is less than or equal to predictive label information resulting from inference*
$$I_{p(\theta |\overset{ˇ}{y})}\;\left[\;p\left(\theta |\overset{ˇ}{y}\right)\;∥\;p\left(\theta \right)\;\right]\leq I_{r(y|\overset{ˇ}{y})}\;\left[\;p\left(y|\overset{ˇ}{y}\right)\;∥\;p\left(y\right)\;\right].$$


This is noteworthy because it tells us that inference always yields a favorable tradeoff between increased model complexity and predictive information. Combining Corollary 14 and Corollary 15, we have upper and lower bounds on model information due to inference
$$I_{p(y|\overset{ˇ}{y})}\;\left[\;p\left(y|\overset{ˇ}{y}\right)\;∥\;p\left(y\right)\;\right]\leq I_{p(\theta |\overset{ˇ}{y})}\;\left[\;p\left(\theta |\overset{ˇ}{y}\right)\;∥\;p\left(\theta \right)\;\right]\leq I_{r(y|\overset{ˇ}{y})}\;\left[\;p\left(y|\overset{ˇ}{y}\right)\;∥\;p\left(y\right)\;\right].$$

### 5.3. Inference Information Constraints

Practitioners of Bayesian inference often struggle when faced with inference problems for models structures that are not well suited to the data. An under-expressive model family is not capable of representing the process being modeled. As a consequence, the posterior collapses to a small set of outcomes that are least inconsistent with the evidence. In contrast, an over-expressive model admits multiple sufficient explanations of the process.

Both model and predictive information measures offer means to understand and address these challenges. By constraining the information gained by inference, we may solve problems associated with model complexity. In this section, we discuss explicit and implicit approaches to enforcing such constraints.

#### 5.3.1. Explicit Information Constraints

Our first approach to encode information constraints is to explicitly solve a distribution that satisfies expected information gained from the prior to the posterior. We examine how information-critical distributions can be constructed from arbitrary states of belief $q_{i}\left(\theta \right)$ for i∈[n]. Again, we may obtain critical distributions with respect to uncertainty by simply setting $q_{0}\left(\theta \right)\equiv 1$. By applying this to an inference, so that n=1 and $q_{1}\left(\theta \right)\equiv p\left(\theta |y\right)$, we recover likelihood annealling as a means to control model information.

**Corollary** **16.**
***Constrained Information.** Given states of belief $q_{i}\left(\theta \right)$ and information constraints $I_{r(\theta )}\;\left[\;q_{i}\left(\theta \right)\;∥\;q_{0}\left(\theta \right)\;\right]=\varphi_{i}$ for i∈[n], the distribution $r^{*}\;\left(\theta \right)$ that satisfies these constraints while minimizing $I_{r(\theta )}\;\left[\;r\left(\theta \right)\;∥\;q_{0}\left(\theta \right)\;\right]$ has the form*
$$r^{*}\;\left(\theta \right)\propto q_{0}\left(\theta \right)\prod_{i=1}^{n}{\left(\frac{q_{i}\left(\theta \right)}{q_{0}\left(\theta \right)}\right)}^{\lambda_{i}}\;for\;some\;\lambda \in R^{n}.$$


**Corollary** **17.**
***Information-Annealed Inference.***
*Annealed belief r(θ) for which information gained from prior to posterior belief is fixed $I_{r(\theta )}\;\left[\;p\left(\theta |\overset{ˇ}{y}\right)\;∥\;p\left(\theta \right)\;\right]=\varphi$ and information $I_{r(\theta )}\;\left[\;r(\theta )\;∥\;p(\theta )\;\right]$ is minimal must take the form*
$$r\left(\theta \right)\propto p{\left(\overset{ˇ}{y}|\theta \right)}^{\lambda }p\left(\theta \right)\;for\;some\;\lambda \in R.$$


Note that the bounds in Section 5.2 still apply if we simply include λ as a fixed model parameter in the definition of the likelihood function so that $p\left(\overset{ˇ}{y}|\theta \right)\mapsto p\left(\overset{ˇ}{y}|\theta ,\lambda \right)\equiv p{\left(\overset{ˇ}{y}|\theta \right)}^{\lambda }$. This prevents the model from learning too much, which may be useful for under-expressive models or for smoothing out the posterior distribution to aid exploration during learning.

#### 5.3.2. Implicit Information Constraints

Our second approach introduces hyper-parameters, λ and ψ, into the Bayesian inference problem, which allows us to define a prior on those hyper-parameters that implicitly encodes information constraints. This approach gives us a way to express how much we believe we can learn from the data and model that we have in hand. Doing so may prevent overconfidence when there are known modeling inadequacies or underconfidence from overly broad priors.

As above, λ parameters influence the likelihood and can be though of as controlling annealing or an embedded stochastic error model. The ψ parameters control the prior on the model parameter θ. For example, these parameters could be the prior mean and co-variance if we assume a Gaussian prior distribution. Therefore, the inference problem takes the form
$$p\left(\theta ,\lambda ,\psi |\overset{ˇ}{y}\right)=\frac{p\left(\overset{ˇ}{y}|\theta ,\lambda \right)p\left(\theta |\psi \right)p\left(\lambda ,\psi \right)}{p(\overset{ˇ}{y})}.$$

In order to encode the information constraints, we must construct the hyper-prior distribution $p\left(\lambda ,\psi \right)=g\left(\varphi_{\theta },\varphi_{y}\right)$ where
$$\begin{array}{cc} \; & \varphi_{\theta }=I_{p(\theta |\overset{ˇ}{y},\lambda ,\psi )}\;\left[\;p\left(\theta |\overset{ˇ}{y},\lambda ,\psi \right)\;∥\;p\left(\theta |\psi \right)\;\right]\;\mathrm{and}\\ \; & \varphi_{y}=I_{r(y|\overset{ˇ}{y})}\;\left[\;p\left(y|\overset{ˇ}{y},\lambda ,\psi \right)\;∥\;p\left(y|\lambda ,\psi \right)\;\right] \end{array}$$
control model information and predictive information, respectively. The function $g\left(\varphi_{\theta },\varphi_{y}\right)$ is the likelihood of λ and ψ given the specified model and prediction complexities. For example, this could be an indicator function as to whether the information gains are within some range. Note that we may also consider other forms of predictive information, such as the predictive generative approximation.

The posterior distribution on model parameters and posterior predictive distribution can be formed by marginalizing over hyper-parameters
$$\begin{array}{cc} \; & p\left(\theta |\overset{ˇ}{y}\right)=\;\int \;d\lambda d\psi \;p\left(\theta ,\lambda ,\psi |\overset{ˇ}{y}\right)\;\mathrm{and}\\ \; & p\left(y|\overset{ˇ}{y}\right)=\;\int \;d\lambda d\psi \;p\left(y|\theta ,\lambda \right)p\left(\theta ,\lambda ,\psi |\overset{ˇ}{y}\right). \end{array}$$

## 6. Negative Information

The possibility of negative information is a unique property of this theory in contrast to divergence measures such as Kullback–Leibler divergence, α-divergences, and *f*-divergences. It provides an easily interpreted notion of whether a belief update is consistent with our best understanding. Negative information can be consistently associated with misinformation in the view of rational belief. That is, if $I_{r(z)}\;\left[\;q_{1}\left(z\right)\;∥\;q_{0}\left(z\right)\;\right]$ is negative, then $q_{0}\left(z\right)$ is a better approximation of rational belief than $q_{1}\left(z\right)$. The consistency of this interpretation would be violated if we could construct $q_{1}\left(z\right)$ that is unambigously better than $q_{0}\left(z\right)$ at approximating r(z). Corollary 11 shows, however, that this is not possible. If we construct $q_{1}\left(z\right)$ by integrating perturbations from $q_{0}\left(z\right)$ that drive belief towards r(z) on all measurable subsets, then information must be positive. The following experiments illustrate examples of negative information and motivate its utility.

### 6.1. Negative Information in Continuous Inference

In the following set of experiments we have a latent variable $\theta \in R^{2}$, which is distributed as N(θ|0,I). Each sample $y^{\left(j\right)}\in R^{2}$ corresponds to realization of an independent latent variable $x^{\left(j\right)}\in R^{2}$ so that $y^{\left(j\right)}=\theta +x^{\left(j\right)}$. Each $x^{\left(j\right)}$ is distributed as $N(x^{\left(j\right)}|0,\sigma_{1}^{2}I)$ where $\sigma_{1}=1/2$. Both prior belief in plausible values of θ and prior predictive belief in plausible values of *y* are visualized in Figure 4. Deciles separate annuli of probability 1/10. The model information we expect to gain by observing 10 samples of *y*, which is also mutual information from Corollary 8, is $I_{p(y,\theta )}\;\left[\;p(y,\theta )\;∥\;p(y)p(\theta )\;\right]=5.36$ bits. See Appendix C for details.

The first observation consists of 10 samples of *y* followed by inference of θ. Subsequent observations each add another 10, 20, and 40 samples respectively. A typical inference sequence is shown in Figure 5. Model information gained by inference from the first observation in the same view is 5.72 bits. As additional observations become available the model information provided by first inference is eventually refined to 5.10 bits. Typically, the region of plausible models θ resulting from each inference is consistent with what was previously considered plausible.

By running one million independent experiments, we constructed a histogram of the model information provided by first inference in subsequent views. That is shown in Figure 6. As a consequence of Postulate 4, the model information provided by first inference must always be positive before any additional observations are made. The change in model covariance in this experiment provides a stronger lower bound of 3.95 bits after first inference, which can be seen in the first view on the left. This bound is saturated in the limit when the inferred mean is unchanged. Additional observations may indicate that the first inference was less informative than initially believed. We may regard the rare cases showing negative information as being misinformed after first inference. The true value of the model θ may be known to arbitrary precision if we collect enough observations. Said value is the realization limit on the right. Under this experimental design, this limit converges to the Laplace distribution centered at mutual information $L(\mu ,{\left(\log2\right)}^{-1})$ computed in the prior view.

From these million experiments, we can select the most unusual cases for which the information provided by first inference is later found to an extreme. Figure 7 visualizes the experiment for which the information provided by first inference is found to be the minimum after observing 160 total samples from the generative process. Although model information assessed following the first observation is a fairly typical value, additional samples quickly show that the first samples were unusual. This becomes highly apparent in the fourth view, which includes 80 samples in total.

Figure 8 visualizes the complementary case in which we select the experiment for which the information provided by first inference is later found to be the maximum. The explantory characteristic of this experiment is the rare value that the true model has taken. High information in inference shows a high degree of unexpected content, given what the prior distribution deemed plausible. Each inference indicates a range of plausible values of θ that is quite distant from the plausible region indicated by prior belief. The change in belief due to first inference is confirmed by additional data in fourth inference.

Finally, we examine a scenario in which the first 10 samples are generated from a different process than subsequent samples. We proceed with inference as before and assume a single generative process, despite the fact that this assumption is actually false. Figure 9 shows the resulting inference sequence. After first inference, nothing appears unusual because there is no data that would contradict inferred belief. As soon as additional data become available, however, information in first inference becomes conspicuously negative. Note that the *one-in-a-million* genuine experiment exhibiting minimum information, Figure 7, gives −11.53 bits after 70 additional samples. In contrast, this experiment yields −47.91 bits after only 10 additional samples.

By comparing this result to the information distribution in the realization limit, we see that the probability of a genuine experiment exhibiting information this negative would be less than $2^{-155}$. This shows how highly negative information may flag anomalous data. We explore this further in the next section.

### 6.2. Negative Information in MNIST Model with Mislabeled Data

We also explored predictive label information in machine learning models by constructing a small neural network to predict MNIST digits [5]. This model was trained with 50,000 images with genuine labels. Training was halted using cross-validation from 10,000 images that also had genuine labels. To investigate how predictive label information serves as an indicator of prediction accuracy, we randomly mislabeled a fraction of unseen cases. Prediction information was observed on 10,000 images for which 50% had been randomly relabeled, which resulted in 5521 original labels and 4479 mismatched labels.

The resulting distribution of information outcomes is plotted in Figure 10, which shows a dramatic difference between genuine labels and mislabeled cases. In all cases, prediction information is quantified from the uninformed probabilities $q_{0}\left(y=y_{i}\right)=1/10$ for all outcomes i∈[10] to model predictions, which are conditioned on the image input $q_{1}\left(y|x\right)$, in the view of the label $r(y|y_{i})$. Total label information—the sum of predictive label information and residual label information: $$I_{r(y|y_{i})}\;\left[\;r\left(y|y_{i}\right)\;∥\;q_{0}\left(y\right)\;\right]=I_{r(y|y_{i})}\;\left[\;q_{1}\left(y|x\right)\;∥\;q_{0}\left(y\right)\;\right]+I_{r(y|y_{i})}\;\left[\;r\left(y|y_{i}\right)\;∥\;q_{1}\left(y|x\right)\;\right]=\log_{2}\left(10\right)\;\mathrm{bits},$$
or roughly 3.32 bits for each case. Both Figure 11 and Figure 12 show different forms of anomaly detection using negative information.

Figure 11 shows that genuine labels may exhibit negative information when predictions are poor. Only 1.1% of correct cases exhibit negative predictive label information. The distribution mean is 3.2 bits for this set. Notably, the first two images appear to be genuinely mislabeled in the original dataset, which underscores the ability of this technique to detect anomalies.

In contrast, over 99.1% of mislabeled cases exhibit negative information with the distribution mean at −18 bits. The top row of Figure 12 shows that information is most negative when the claimed label is not plausible and model predictions clearly match the image. Similarly to Section 6.1, strongly negative information indicates anomalous data. When incorrect predictions match incorrect labels, however, information can be positive, as shown in the bottom row. The cases appear to share identifiable features with the claim.

## 7. Conclusions

Just as belief matures with accumulation of evidence, we hold that the information associated with a shift in belief must also mature. By formulating principles that articulate how we may regard information as a reasonable expectation that measures change in belief, we derived a theory of information that places existing measures of entropic information in a coherent unified framework. These measures include Shannon’s original description of entropy, cross-entropy, realization information, Kullback–Leibler divergence, and Lindley information (uncertainty difference) due to an experiment.

Moreover, we found other explainable information measures that may be adapted to specific scenarios from first principles, including the log pointwise predictive measure of model accuracy. We derived useful properties of information, including the chain rule of conditional dependence, additivity over belief updates, consistency with respect expected future observations, and expected information in future experiments as mutual information. We also showed how this theory generalizes information-critical probability distributions that are consistent with observed expectations analogous to those of Jaynes. In the context of Bayesian inference, we showed how information constraints recover and illuminate useful annealed inference practices.

We also examined the phenomenon of negative information, which occurs when a more justified point of view, based on a broader body of evidence, indicates that a previous change of belief was misleading. Experiments demonstrated that negative information reveals anomalous cases of inference or anomalous predictions in the context of machine learning.

The primary value of this theoretical framework is the consistent interpretation and corresponding properties of information that guide how it may be assessed in a given context. The property of additivity over belief updates within the present view allows us to partition information in a logically consistent manner. For machine learning algorithms, we see that total information from the uninformed state to a label-informed state is a constant that may be partitioned into the predicted component and the residual component. This insight suggests new approaches to model training, which will be the subject of continuing research.

### Future Work

The challenges we seek to address with this theory relate to real-world applications of inference and machine learning. Although Bayesian inference provides a rigorous foundation for learning, poor choices of prior or likelihood can lead to results that elude or contradict human intuition when analyzed after the fact. This only becomes worse as the scale of learning problems increases, as in deep neural networks, where human intuition cannot catch inconsistencies. Information provides a metric to quantify how well a model is learning that may be useful when structuring learning problems. Some related challenges include:Controlling model complexity in machine learning to avoid memorization;Evaluating the influences of different experiments and data points to identify outliers or poorly supported inferences;Understanding the impact of both the model-structure and fidelity of variational approximations on learnability.

## Figures and Tables

**Figure 1 entropy-22-00108-f001:**
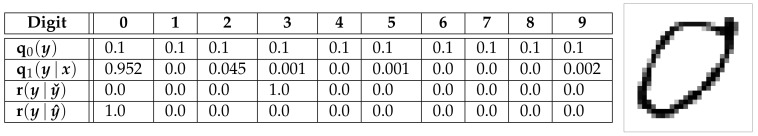
Example of the evolution of plausible labels for an image. Without evidence, the probability distribution $q_{0}\left(y\right)$ assigns equal plausibility to all outcomes. A machine learning model processes the image and produces predictions $q_{1}\left(y|x\right)$. The incorrect label **3** is represented by $r(y|\overset{ˇ}{y})$. After observing the image, shown on the right, the label is corrected to **0** in $r(y|\hat{y})$.

**Figure 2 entropy-22-00108-f002:**
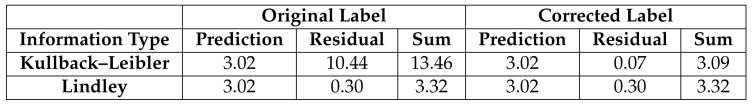
Information measurements before and after label correction. Neither construction of prediction information allows the computation to account for claimed labels. Residual information, however, decreases in the KL construction when the label is corrected. The Lindley forms are totally unaffected by relabeling.

**Figure 3 entropy-22-00108-f003:**
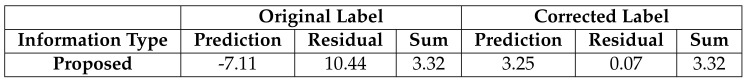
Information measurements using our proposed framework. Total information is a conserved quantity, and when our belief changes, so do the information measurements. Negative prediction information forewarns either potential mislabeling or a poor prediction.

**Figure 4 entropy-22-00108-f004:**
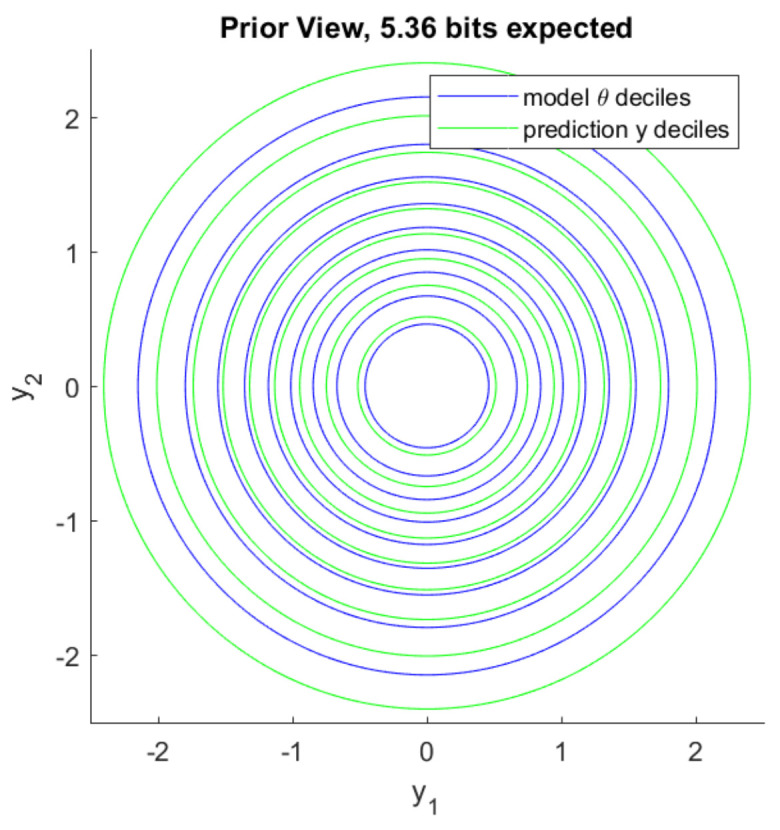
Prior distribution of θ and prior predictive distribution of individual *y* samples. The domain of plausible θ values is large before any observations are made.

**Figure 5 entropy-22-00108-f005:**
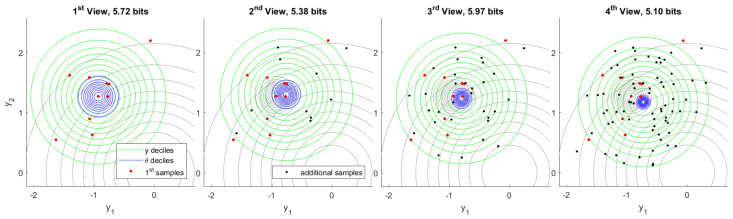
Typical inference of θ from observation of 10 samples of *y* (left) followed by 10, 20, and 40 additional samples, respectively. Both prior belief and first inference deciles of θ are shown in gray. As observations accumulate, the domain of plausible θ values tightens.

**Figure 6 entropy-22-00108-f006:**
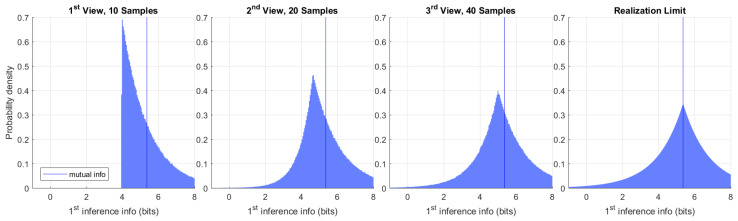
Histogram of the earliest inference information in the observation sequence. The vertical line at 5.36 bits is mutual information. Information is positive after first inference, but may drop with additional observations. The limiting view of infinite samples (realization) is shown on the right.

**Figure 7 entropy-22-00108-f007:**
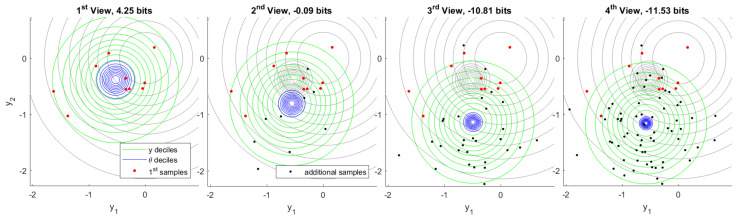
Minimum first inference information out of one million independent experiments. This particularly rare case shows how first samples can mislead inference, which is later corrected by additional observations. The fourth inference (right) bears remarkably little overlap with the first.

**Figure 8 entropy-22-00108-f008:**
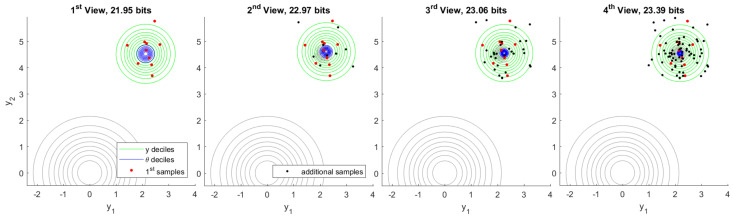
Maximum first inference information out of one million independent experiments. The true value of θ has taken an extremely rare value. As evidence accumulates, plausible ranges of θ confirm the first inference.

**Figure 9 entropy-22-00108-f009:**
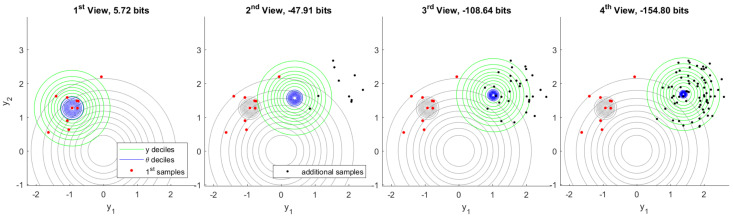
Inconsistent inference. The first 10 samples are drawn from a different ground truth than subsequent samples, but inference proceeds as usual. As additional data become available, first inference information becomes markedly negative.

**Figure 10 entropy-22-00108-f010:**
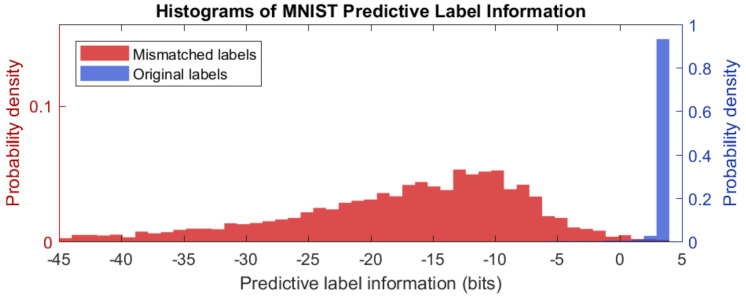
Histogram of information outcomes for mismatched and original labels. Correct label information is highly concentrated at 3.2 bits, which is 95.9% of the total information contained in labels. Mislabeled cases have mean information at –18 bits, and information is negative for 99.15% of mislabeled cases.

**Figure 11 entropy-22-00108-f011:**
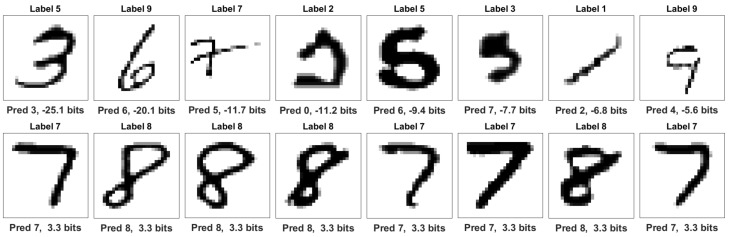
Original MNIST labels. The top row shows lowest predictive label information among original labels. Notably, the two leading images appear to be genuinely mislabeled in the original dataset. Subsequent predictions are poor. The bottom row shows the highest information among original labels. Labels and predictions are consistent in these cases.

**Figure 12 entropy-22-00108-f012:**
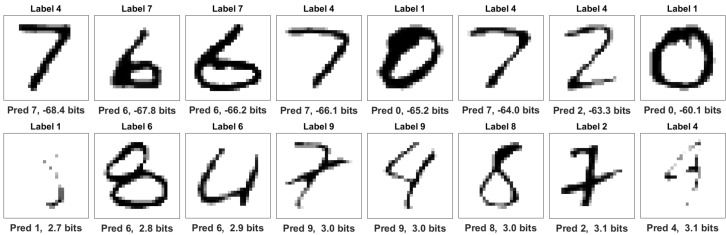
Mislabeled digits. The top row shows the lowest predictive label information among mislabeled cases. In each case, the claimed label is implausible and the prediction is correct. The bottom row shows the highest prediction information among mislabeled cases. Although claimed labels are incorrect, most images share identifiable features with the claim.

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
