# Peer review of "Generalizing Information to the Evolution of Rational Belief"

_entropy, 2020, doi:10.3390/e22010108_

Round 1

Reviewer 1 Report

This is a contribution to the mathematical theory of rational belief change. The authors write from the perspective and with some of the terminology of the field of machine learning. The subject of their paper is however very general, and applies to generic situations in which states of belief are updated. This mismatch between the specific viewpoint of the authors (machine learning) and the generality of the issues they address is perhaps responsible for the lack of accessibility of the paper. The authors sometimes use terms of their own, it seems, in places where there already exists accepted usage of other terms (e.g., in probability theory). Moreover, they are often not clear in their explanations. I also found the structure of the paper rather opaque: concepts are used without explanation in the beginning and readers are referred to later sections if they want to know the meaning of these concepts. 

Nevertheless, I think the mathematical parts may be useful and, I think,  publishable.  I would like to urge the authors to revise their explanations of what their proposals are about. In particular, they should be clear in their introduction and they should make this introduction self-contained.

Examples of points to take into account.

On page 3 the distribution r(x) is introduced as "representing training labels". I found the meaning of this obscure. It does not help that the authors refer to later sections for "details regarding notation". They should make it completely clear here, in their introduction, what the concrete situation is that they have in mind, and what the rationale is for the introduction of r(x) [perhaps representing the actual state of affairs?]. 

Also in section 1, and further on in the manuscript, the authors talk a lot about "rational" belief, but they never provide a definition. In Bayesianism it is standard to reserve this term (rationality) for belief and belief changes that are not vulnerable to Dutch book arguments, but it seems that the authors have something very different in mind. But I could not find out the details. Why is section 2.2 called "Foundations of Reason"?

As a general mathematical theory of probabilistic inference, the scheme proposed by the authors should of course be insensitive to whether the probabilities are objective or epistemic. But what they write about this on page 4 is not really correct. Bell did not show that quantum probabilities are "intrinsic to QM" (an example of the vague use of terms in the article; I take it that "objective" is meant). He showed that epistemic interpretations of quantum probabilities must involve features of nonlocality. The second example, statistical mechanics, is also incorrect. The probabilities in SM are usually taken to be epistemic (in classical SM, that is).  

In 2.4 there is the announced explanation concerning the meaning of r(x). But now this r(x) first turns out to be an arbitrary state of belief, in seeming contradiction with what was mentioned in the introduction. And a little bit further on in the same section we are surplussed to find that r(z) represents the state of "rational belief".  This is defined as "belief consistent with all the available evidence". I believe that q0 and q1 should also be consistent with all available evidence. "Consistence" with all evidence is a very weak demand. Anyway, the use of "rational belief" in this section seems to contradict what is usual in Bayesian approaches. The authors should be clearer about what they have in mind. A concrete example would certainly help.

The way the authors use the distinction between objective and subjective probabilities also seems non-standard. Usually, objective probabilities are meant to represent probabilities in nature (e.g., quantum probabilities according to the Copenhagen interpretation), whereas subjective probabilities represent our belief and our willingness to place bets. 

In general, in the later sections, the precise motivation for considering the third perspective r remained rather murky to me. Who is the rational agent that works with this state of belief? If r is rational, does that mean that q0 and q1 (conditioned on the evidence) are not rational? That would be an important break with the very core of Bayesianism. 

Summing up, I feel that the details of the proposed scheme and its concrete motivation should become clearer before publication is warranted. 

Reviewer 2 Report

The paper describes a general scheme where the information is considered as the update of belief regarding some point of view. This quantity is axiomatized and the form of the information function is deduced. This generalized KL divergence, and other information measures. They exemplify the approach on several examples. The paper is potentially very interesting with nice results, but several points must be improved/updated/added:

The theory is very nice since it includes all well-known measures, but my question is whether we get any real merit from it, i.e., if there are some applications where KL divergence, etc. fail and this form gives us better results. Please clarify this in a separate section. Postulate 1 corresponds to the well-known trace form of entropies. What if we allow some more generalized form, as e.g., sum-class form (a function of this). You can maybe relate this axiom to the Shore-Johnson axiom 3 (see e.g., 10.1103/PhysRevLett.122.120601). Maybe the axioms can be also interpreted through the updating rules, please give some connection to the theory of statistical inference For L^p norms: please explain why there is the absolute value in the definition, this makes it a bit difficult to interpret.  Sections 4 and 5 are a bit redundant, or they can be rewritten in much compact form (maybe a table or so) Section 7 is really not clear to me. Please explain more clearly what is the interpretation of the negative information and what are their applications. You mention the example with the mislabeled data but this is not so illustrative. Maybe write explicitly an example of r,q0, and q1 for which we get negative information Future work: Can you comment on what happens if you consider correlations in your system? Will you also consider some entropic functionals due to Rényi in that case? How would the postulates change? Literature: the paper is based on famous papers from information theory, but I miss some new references showing the current state-of-the-art. Please add some recent related works on this topic to the introduction.

Round 2

Reviewer 1 Report

I think that the manuscript has improved a lot, so that it now has become possible to understand what the work is aiming at. I still find the presentation less than crystal clear, especially from a conceptual point of view. But I expect that the paper will most of all be of interest to a specialized audience of experts in machine learning, and that for these readers the technical details of the proposal will be the important part. The authors clearly write from the perspective of their own specialty and I think that they should not be forced to write an essay in philosophical logic/ probability. I therefore think the paper should be published as is.